# The Application of X-ray Micro-CT in the Study of HTS Tape Coils

**Vitaly B. Minasyan** [1,*] **, Nikolay S. Ivanov** [2] **, Elizaveta A. Malykh** [1] **, Yuri A. Zanegin** [2] **and Bruno Douine** [3]

[1] Digital Technologies and Information Systems Department, Moscow Aviation Institute (The National Research University), 125993 Moscow, Russia; malyhfistina@gmail.com

[2] Electrical Power, Electromechanics and Biotechnical Systems Department, Moscow Aviation Institute (The National Research University), 125993 Moscow, Russia; ivanovns@mai.ru (N.S.I.); zaneginsy@mai.ru (Y.A.Z.)

[3] GREEN, University of Lorraine, F-54000 Nancy, France; bruno.douine@univ-lorraine.fr

* Correspondence: minasyanvb@mai.ru

**Abstract:** In the process of manufacturing products from high-temperature superconductors (HTS), quality control must be carried out. Traditionally, for HTS coils, electrical tests are carried out to determine critical current. In the case of an unacceptable result, it is necessary to determine the cause. Therefore, it is necessary to develop nondestructive testing methods. This article proposes a technology for manufacturing quality evaluation. It is based on determining the actual location of the tape and the gaps between the turns and rows of the coil and analyzing these values. For this purpose, samples were scanned using computed tomography (CT) with a Nordson Dage XD7600NT X-ray inspection system with a μCT module. The obtained data were analyzed using VolumeGraphics VGStudio 2.2 software. Furthermore, the proposed technology can be used as part of a predictive analysis of the state of HTS coils in the windings of electrical machines.

**Keywords:** high-temperature superconductor; racetrack coils; high-temperature superconductor coil; nondestructive testing; predictive analysis; X-ray; computed tomography; X-ray micro-computed tomography; 3D voxel image

## 1. Introduction

Modern second-generation high-temperature superconductors (HTS) are composite tapes consisting of a large number of layers with different physical properties [1]. It is known that a lot of experimental coils are produced in different research centers [2–9]. However, the manufacturing technology for HTS tapes is still improving, which has led to a change in their properties. In particular, the value of critical current has increased and its field dependency has become stronger, in addition to the tape thickness having decreased [10]. Due to the fact that the parameters of HTS tapes have changed, it has become necessary to create a new technology for manufacturing HTS coils, which would take into account the features of modern materials. An important stage in the development of technology, as well as its subsequent implementation, is the quality control of manufacturing, including nondestructive methods.

Traditionally, electrical methods are used for testing products made from HTS tape [11]. However, in some cases, especially when the critical current is below the required value, it is necessary to determine the cause of this decrease. For these purposes, nondestructive flaw-detection methods should be developed that will allow for a conclusion to be drawn about the quality of the coil manufacturing. In addition, nondestructive testing methods can also be used during long-term operation of HTS coils, when it is necessary to carry out routine maintenance and determine the state of the coils for the purpose of predictive analysis.

The manufacture of racetrack coils with characteristics lower than expected may be associated with mechanical damage to the tape during winding, thermomechanical stresses during coil cooling, insufficient adhesion of the turns to each other and the support using a compound.

This article proposes a technology for studying HTS coils, which makes it possible to evaluate the quality of manufacture based on determining the actual location of the tape and the gaps between the turns and rows of the coil and analyzing these values, while electrical testing gives the opportunity to determine only the functional state of the coil—whether it is in superconducting state or not. For this, three samples were produced and scanned using computed tomography (CT) with a Nordson Dage XD7600NT X-ray Inspection System with a µCT module. X-ray micro-CT inspection is considered as nondestructive [12,13], since the CT scan is performed on the entire coil, after which it remains intact and afunctional. It is known that X-rays could provide a negative influence on the HTS layer in tape [14]. To examine this, in the first step short samples were tested to determine the influence of X-rays on HTS tape critical current. According to the test results, there was no degradation of critical current. Despite this, research of such influence will be investigated in future projects. The obtained data were processed using VolumeGraphics VGStudio 2.2 software.

## 2. Materials and Methods

### 2.1. Research Object

The HTS coil (Figure 1) has the shape of a racetrack and is made using double-pancake technology, i.e., consisting of two rows of tape wound in different directions. The transition between the rows of tape is located inside of the coil.

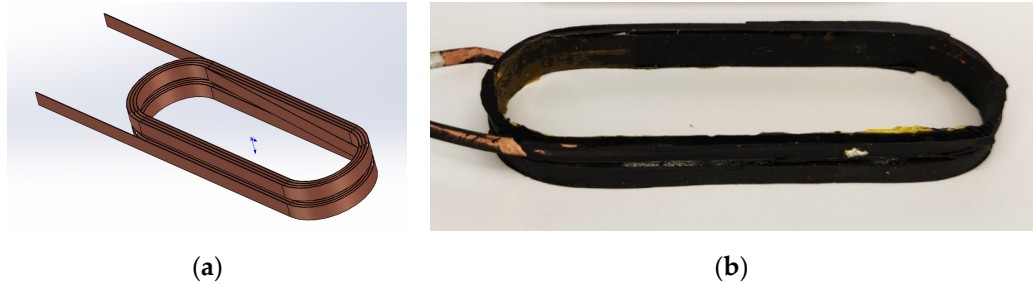

(**a**)                                                 (**b**)

**Figure 1.** HTS coil. (**a**) Model; (**b**) appearance of the sample.

Considered coils are made of HTS tape manufactured by SuperOX with Loctite Stycast [15] compound. The parameters of the used tape and coils are given in Table 1 [1]. Figure 1b shows the appearance of the coil. All samples were first tested in liquid nitrogen (77K) to determine critical current. Then, CT scanning was performed at room temperature. The design of the used µCT module does not allow scanning of the sample placed in liquid nitrogen.

**Table 1.** Parameters of the HTS tape and coils under study.

| HTS Tape Manufacturer | SuperOX, 4 mm, Substrate 60 µm, Copper 5 µm |
|:---:|:---:|
| Insulation | Polyimide |
| Coil type | Double pancake |
| Number of turns | 12 |
| Axial length, mm | 100 |
| **Critical current at 77K, A:** | |
| Short sample in own field | 150 |
| Coil #1 | 93 |
| Coil #2 | 95 |
| Coil #3 | 97 |

## 2.2. Research Methodology

When choosing a technology for studying HTS coils, X-ray imaging limitations should be taken into account. Two-dimensional systems based on an X-ray tube with a fan-beam or cone-beam radiation flux cannot provide sharp images of certain object shapes. Since the ratio between the width of the tape and the gap between the turns is quite large, X-rays on their way to the detector pass through several turns of the tape, which leads to blurring of the edges of the tape and perspective distortions in the resulting images [16], which is schematically shown in Figure 2a. The presence of two rows of tape in the coil makes this problem even more acute, and as a result, X-ray images do not allow us to quantitatively determine the size of the gaps between the turns, and in some cases, determine the presence of the gap itself (Figure 2b,c). Therefore, the study of the gaps between the turns of the tape on two-dimensional x-ray images is not appropriate.

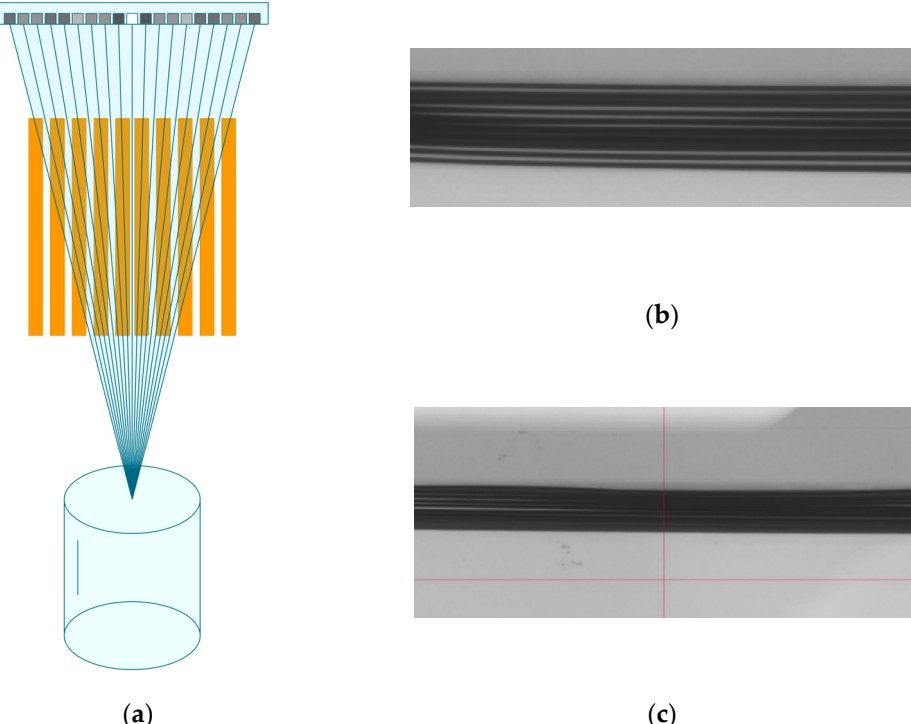

(a)  (b)  (c)

**Figure 2.** X-rays of HTS coils. (**a**) Scheme of the image; (**b**,**c**) examples of X-rays.

To solve this problem, the use of CT is proposed [17]. This imaging technique consists of successive shooting of an object with the object's rotation [18], after which a three-dimensional raster image is reconstructed using mathematical transformations [19,20]. A voxel is a three-dimensional raster unit, an analogue of a two-dimensional pixel in three-dimensional space, and represents a cube. The image consists of many of the cubes (Figure 3) [21]. The voxel dimensions are calculated by the software during image reconstruction, which makes it possible to measure distances in any plane.

In this work, scanning of samples using the method of computed tomography was performed on a Nordson Dage XD7600NT X-ray inspection system with a μCT module at X-ray tube parameters of 160 kV, 10.0 W. Processing of the received data is carried out using VolumeGraphics VGStudio 2.2 software. It should be noted that the size of the resulting tomogram is equal to 1024 × 1024 × 1024 voxels, and the resolution in microns depends on the geometric magnification during scanning, which is limited by the maximum rotation radius of the mounted sample. Since the center of the obtained images (and, as a result, of the tomograms) is located on the rotation axis, the sample must be placed in such a way that the region of interest would be located on the rotation axis. The maximum achievable magnification is limited by the maximum radius of rotation of the fixed sample and can

reach up to 3 μm on the used setup. However, the dimensions of the HTS tapes limit the resulting resolution to 12 μm.

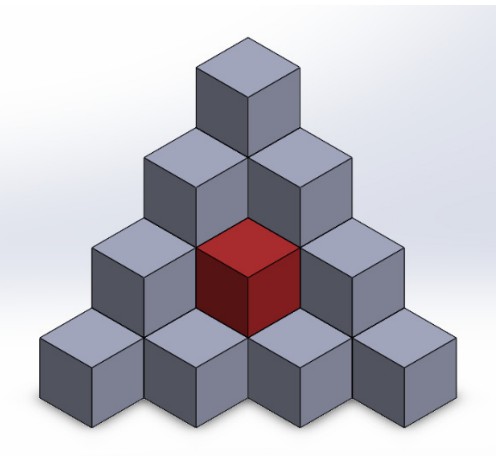

**Figure 3.** An example of a voxel model, one of the voxels is highlighted in color.

For CT scanning, a special HTS coil holder has been developed for fixing it on the rotary axis of the X-ray setup (Figure 4). This coil holder must provide sufficient rigidity along the axis of rotation, since with a coil length of about 100 mm, gravity can cause coil sagging, and as a result, misalignment during CT reconstruction. To obtain a high-quality image that allows one to estimate the distance between the turns, the required resolution of the tomogram should be at least $15 \times 15 \times 15$ μm, which is due to the size of the HTS tape used (100 μm). In accordance with this, in the developed holder, one side of the coil is placed on the axis of rotation. This arrangement of the coil makes it possible to use a sufficient geometric magnification and an actual resolution of $12 \times 15 \times 15$ μm, which makes it possible to determine the gaps between the turns.

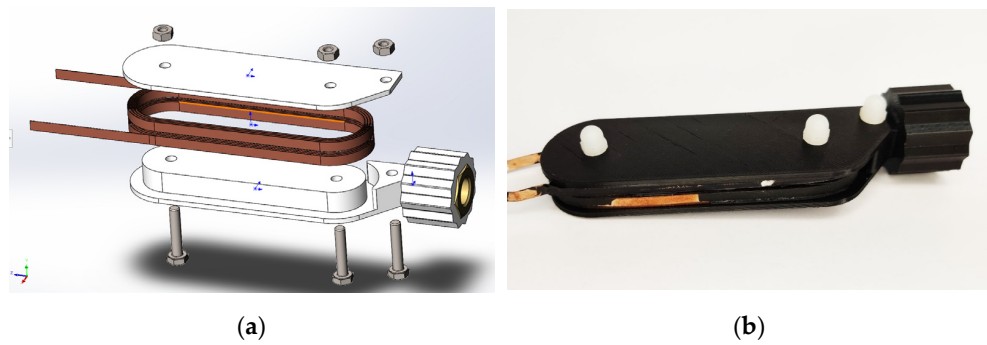

(**a**)                               (**b**)

**Figure 4.** HTS coil in holder. (**a**) Model; (**b**) appearance of the sample.

The result of the computed tomography method is a tomogram—a three-dimensional voxel image with a resolution of $1024 \times 1024 \times 1024$ voxels (Figure 5a).

It is supposed that data processing is carried out along the rotation axis. To do this, the data are presented as a set of 1024 slice images with a resolution of $1024 \times 1024$ pixels, where each of the images is a slice of the coil (Figure 5b). A tomogram with such a resolution displays a space of $12.3 \times 15.6 \times 15.6$ mm in size, which contains a section of the coil. To obtain data on the entire linear part of the coil and ensure that the received data overlap by 2 mm, it is necessary to study 9 areas for each of the sides. Thus, for the coil, $1024 \times 9 \times 2$ = 18,432 slice images are obtained. Since the processing of such a volume of data can take a significant time and there is no need to measure the gaps every 12 μm, it was decided to

measure the gaps in 1 mm steps, which corresponds to 13 slices (positions at 0, 1, 2, . . . , 12 mm) on each tomogram and $13 \times 9 \times 2 = 234$ slices in total.

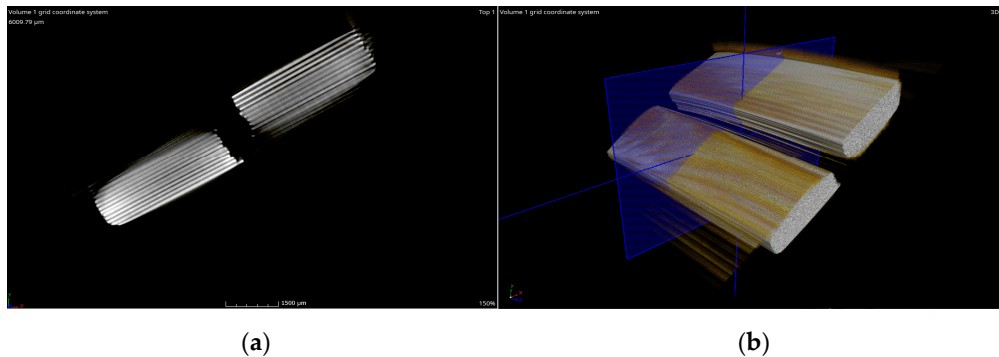

<div align="center">(<b>a</b>)                          (<b>b</b>)</div>

**Figure 5.** A tomogram slice of the HTS coil (**a**) and its location on the volumetric image (**b**).

Figure 6 schematically shows the structure of the coil and measurement locations, and Figure 7 is an example image of a coil section obtained during a CT scan, indicating measurement locations. Table 2 shows a comparison of positions in the diagram of Figure 6 and an example of the resulting image in Figure 7. All measurements are carried out only in the linear part of the coil. In this case, the reference point on the graphs corresponds to point 0 in Figure 6, the direction of the coil bypass is also shown in Figure 6 by arrows.

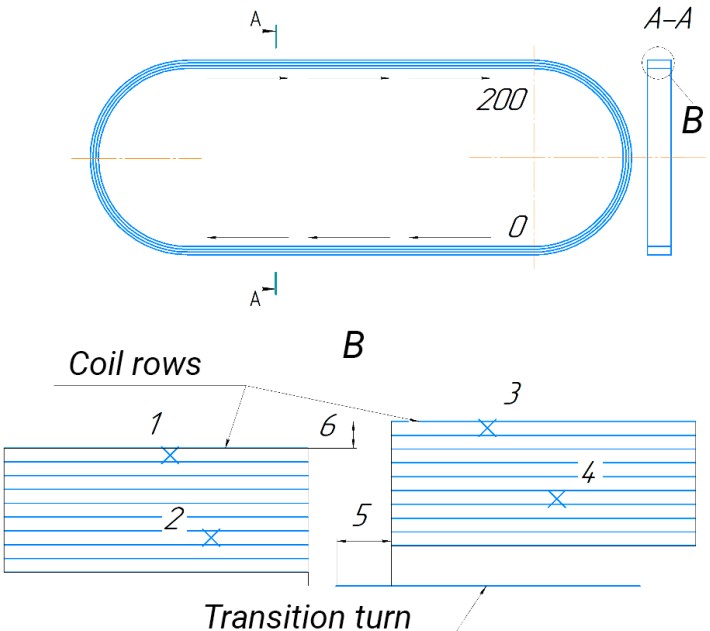

**Figure 6.** Layout of measurements on slices of tomograms. In view B, the lines show HTS tapes; the distances between them are increased for clarity.

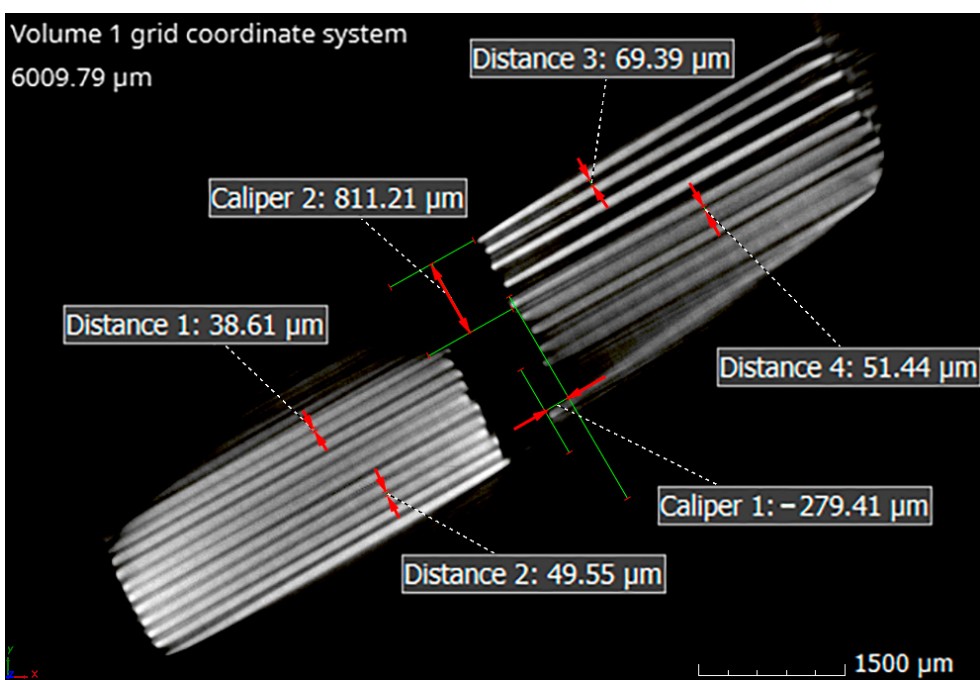

**Figure 7.** Slice of a tomogram with the location of measuring points.

**Table 2.** Designation of measurement points in Figures 6 and 7.

| Description | Designation in Figure 6 | Designation in Figure 7 |
|---|---|---|
| Places for measuring the distance between adjacent turns in different parts of the coil | 1, 2, 3, 4 | «Distance 1», «Distance 2», «Distance 3», «Distance 4» |
| Place for measuring the position of the transition turn relative to one side of the coil | 5 | «Caliper 1» |
| Place for measuring the relative distance between the two layers of the coil | 6 | «Caliper 2» |

### 3. Results

*3.1. Graphs of Gaps between Turns of the Coil*

As a result of the measurements, graphs showing the numerical values of the gaps between the turns in different areas of the coils and the location of the transition turn were obtained.

Figure 8 shows graphs of gap values between the turns at measurement points 1, 2, 3 and 4 in accordance with the scheme of Figure 6. Figure 9 shows graphs of the average value and standard deviation of measurements according to the four points.

The graphs on Figure 8 are significantly inhomogeneous. The distances between the turns at different measurement points in some cases differ significantly. Such a spread in the values of the gaps may be the result of insufficient adjustment of the winding equipment and/or uneven tape tension. However, the maximum distance does not exceed 120 µm, which indicates a fill factor close to 0.5 considering the 0.1 mm HTS tape thickness. A rapid change in the average value (Figure 9) may be due to unstable tape tension, inhomogeneity in the used compound, insufficient wettability of the copper surface of the tape, and volumetric inclusions between layers. In general, in the case of good winding, the graphs in Figures 8 and 9 should be close to a straight line. At the same time, in the process of developing the technology, the permissible values of the gaps, as well as their standard deviation, should be formulated, which will be controlled during the CT examination of the manufactured samples.

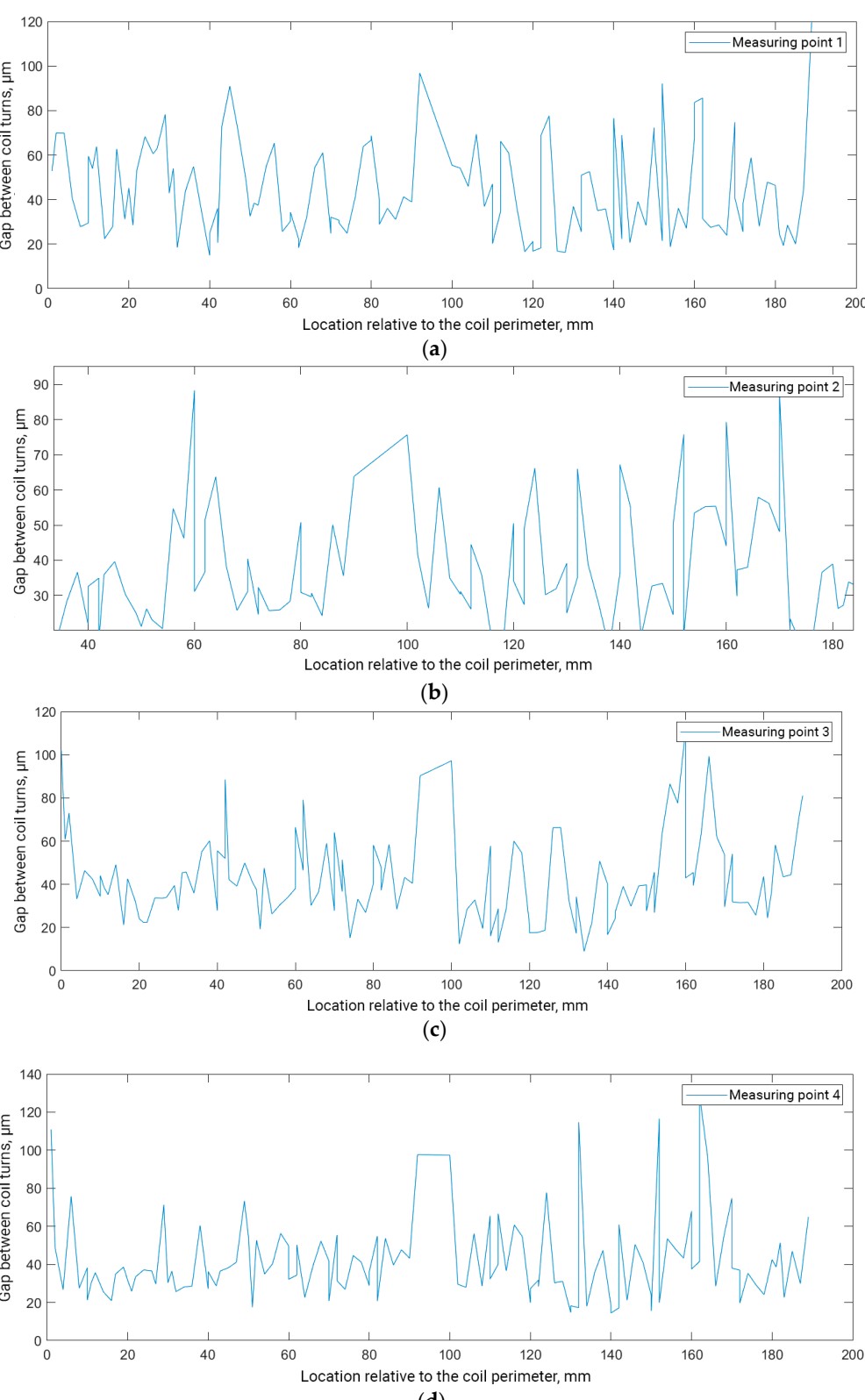

**Figure 8.** Graphs of gap values between turns for coil No. 1, measured at points 1 (**a**), 2 (**b**), 3 (**c**), 4 (**d**) in accordance with Figure 6.

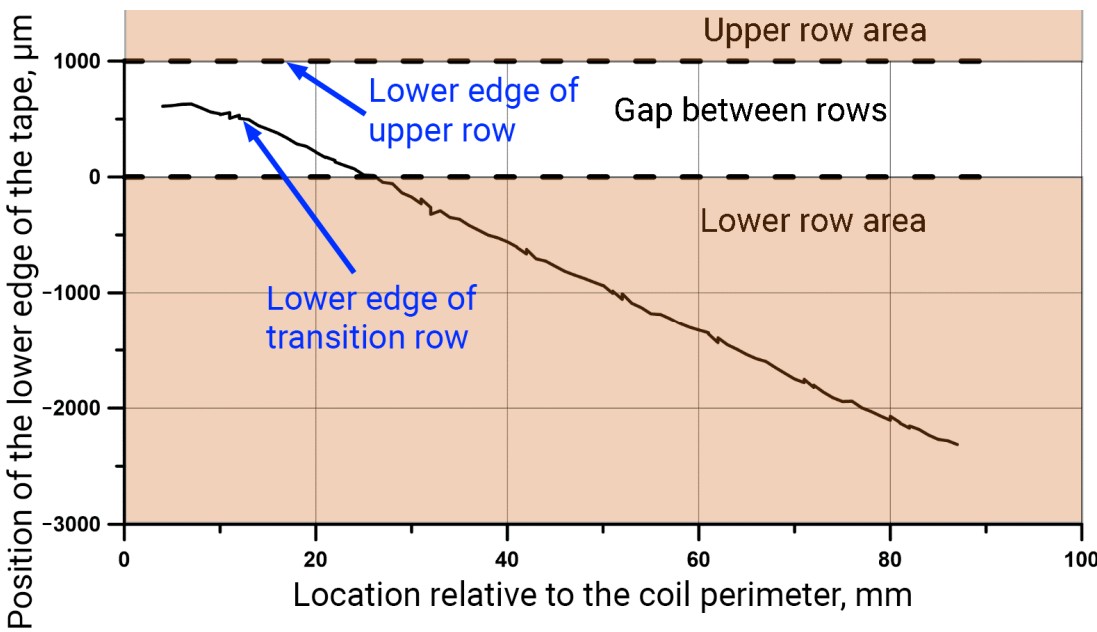

**Figure 9.** Coil #1 transition turn location.

### 3.2. Graphs of the Location of the Transition Turn

One of the indicators of winding quality is the location of the transition turn, because its excessive bending can lead to deformation of the tape and destruction of the superconducting layer. Figure 9 shows the position of the lower edge of the transition turn versus the axial length of the coil. The origin in the graph of Figure 9 coincides with the upper edge of the lower row of the coil. In Figure 6, this distance is indicated by the number 5, in Figure 7—"Caliper 1". The positive values in Figure 9 refer to the gap between the turns, and the negative values refer to the area of the bottom layer of the coil. In the range from 0 mm to 4 mm, measurements could not be carried out due to technical difficulties in reliability of determining the position of the transition coil.

Based on the data in Figure 9, the position of the transition turn can be determined and compared with the ideal case of stacking the HTS tape. In the manufacture of HTS coils of similar shape and size, the position of the turn should be the same. A deviation in any area will indicate a violation of manufacturing technology.

### 4. Discussion

The analysis of these data can be used to create methods for quality control of coil winding. The size of the gaps between the turns of the coil can serve as a marker for assessing the quality and stability of the technological operation. The smaller and more uniform the gap of this value is along the coil length, the better the winding technological operation is performed. In general, it is advisable to conduct a CT study of each produced HTS coil to assess the uniformity of winding. However, the radiodensity of the used compound does not allow one to determine the thickness of its layer between the turns, as well as the presence of voids in it. The presence of randomly located voids (volume without compound) can lead to a complexly profiled vibration of the turns in AC coils. External factors could also produce vibrations in the case of presence of voids. It will lead to an additional change in the pattern of internal stresses, both in the tape itself and between its turns. This problem can be solved by a systematic study of HTS coils during their operation and by comparing the gap values. An increased gap value will indicate insufficient fixation of the turns to each other by compound. This analysis will be a part of the predictive analytics of the state of HTS coils and failure prediction. Thus, the proposed CT scanning technique can be taken as a basis for the analysis of the state and the need

for maintenance of HTS coils operating in electric machines, magnetic systems of MRI machines, electromagnets, etc.

## 5. Conclusions

A method for nondestructive quality control of the manufacturing of HTS coils is presented in this paper. This method allows for the obtaining of values of gaps between the turns, the location of the transition turn and the distance between the rows of the coil. X-ray micro-CT scanning was performed at room temperature without applied or pinned magnetic fields and currents. In particular, for the coils it was established that:

1. For the studied coils, the interturn distances do not exceed 120 μm, which with a tape thickness of 100 μm corresponds to a fill factor of 0.5;
2. Nonuniformity of the turn-to-turn gap may be caused by the uneven capture of the compound on the tape surface, which may occur due to insufficient wettability of the copper coating of the tape;
3. In general, the graphs of the distance between the turns should be close to a straight line. At the same time, in the process of developing the technology, the permissible values of the gaps, as well as their standard deviations, should be formulated. The values will be controlled during the CT examination of the manufactured samples;
4. The location of the transition turn is estimated in the comparison with the ideal position. For the three coils under study, some deviation from the ideal position is observed. A significant deviation can serve as an indicator of coil deformation and damaged HTS layer.

**Author Contributions:** Methodology, analysis, V.B.M. and N.S.I.; investigation, V.B.M., Y.A.Z. and E.A.M.; visualization: V.B.M. and E.A.M.; supervision, B.D.; writing—original draft preparation, V.B.M.; writing—review and editing, N.S.I. All authors have read and agreed to the published version of the manuscript.

**Funding:** The study was carried out with the financial support of a project by the Russian Federation represented by the Ministry of Education and Science of the Russian Federation, agreement No. 075-15-2020-770.

**Institutional Review Board Statement:** Not applicable.

**Informed Consent Statement:** Not applicable.

**Data Availability Statement:** Not applicable.

**Conflicts of Interest:** The authors declare no conflict of interest.

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
