# Peer review of "The Application of X-ray Micro-CT in the Study of HTS Tape Coils"

_inventions, doi:10.3390/inventions7030060_

Round 1

Reviewer 1 Report

The following comments and suggestions are given for authors: 

1-Remove acronyms from keywords, such as HTS and CT. Some additional keywords can also be added related with the presented subject. As an example for better introdution of the theme, the list of keywords could be: high-temperature superconductor; racetrack coils; coil rows; transition turn; gap between turns; non-destructive testing; predictive analysis; X-Ray micro-computed tomography; 3D voxel image.

2-The following reference from MDPI-Electronics journal, should be added to reinforce the second sentence of the introdution: "It is known that a lot of experimental coils were produced in different research centers [2–9]."

[9] Li Lu, Wei Wu, Xin Yu, and Zhijian Jin. High-Temperature Superconducting Non-Insulation Closed-Loop Coils for Electro-Dynamic Suspension System. MDPI-Electronics 10(16), 1980; 2021.

3-Clarify with a sentence what are the admissible testing conditions. If the X-Ray micro-CT quality control testing: i) Could be done with the coils cooled at cryogenic temperatures and under effect or not of applyed or pinned magnetic fields and currents; or ii) Should be done with non-cooled coils at the room temperature.

4-Explain better to which extend the X-Ray micro-CT testing is considered a non-destructive method when compared with other traditional electric testing methods.

5-Mention if the application of X-Rays could ionize nanoparticles from the tape layers and with this generate currents or damage the structure of tape layers. Refer to the following article from MDPI-nanomaterials:

Filip Antoncík, Ondrej Jankovský, Tomáš Hlásek, and Vilém Bartunek. Nanosized Pinning Centers in the Rare Earth-Barium-Copper-Oxide Thin-Film Superconductors. MDPI-Nanomaterials 10(8), 1429; 2020.

6-Explain more clearly how did you calculate 234 slices in line 119.

7-Increase the size of text fonts in the labels of Figure 7, to become more legible.

8-Figure 9 needs axis titles and units. Also include more indication labels for the reader to indentify better reference positions of the transition turn.

9-From line 169 to line 176, the authors indicate Figure 10, but there are only nine figures. I suppose they should reder instead to Figure 9.

10-In the Conclusions refer to the conditions of application of X-Ray micro-CT testing, if with the coils cooled at cryogenic temperatures and if under applyed or pinned magnetic fields and currents.

Author Response

Thank you for your questions and comments, we have prepared revision in accordance with your comments. Please see the attachment.

Reviewer 2 Report

The manuscript presented a very interesting non-destructive method to measure and evaluate the positioning of tape conductors inside a racetrack coil. The CT method provides a significant insight into the superconducting coils and magnets. These insights can help understand the magnet fabrication and performance. I recommend the publication of the paper that can be of great interest to the magnet community. 

A few comments and questions:

- Figure 9 seems missing. The figure labelled as Fig. 9 seems to be Fig. 10. Please double check. 

- More relevant references on the application of CT on superconducting magnets can benefit the paper. A few examples:

-- https://link.springer.com/chapter/10.1007/978-1-4615-2439-7_165

-- https://doi.org/10.1038/s41598-021-01999-5

-- https://doi.org/10.1038/s41598-021-87475-6

-- https://doi.org/10.1103/PhysRevSTAB.18.061002  

- Was the Stycast 2850 applied to the coil via vacuum-pressure impregnation? Do you observe any voids in the epoxy from the CT images?

- What can be improved in the reported CT method? What is the limitation on the dimensions of the sample that can be measured?  

Author Response

(The authors gave the same response as above.)

Round 2

Reviewer 1 Report

The authors complied to all the reviewer observations.

Additional sugestion:

To reduce complexity please remove the term "high-temperature superconductor coil" from the keywords. This is because you already have the terms "high-temperature superconductor", and "Racetrack coil" to be kept.